# Revising the Classic Computing Paradigm and Its Technological Implementations

János Végh [ID]

Kalimános BT, 4032 Debrecen, Hungary; Vegh.Janos@gmail.com

**Abstract:** Today's computing is based on the classic paradigm proposed by John von Neumann, three-quarters of a century ago. That paradigm, however, was justified for (the timing relations of) vacuum tubes only. The technological development invalidated the classic paradigm (but not the model!). It led to catastrophic performance losses in computing systems, from the operating gate level to large networks, including the neuromorphic ones. The model is perfect, but the paradigm is applied outside of its range of validity. The classic paradigm is completed here by providing the "procedure" missing from the "First Draft" that enables computing science to work with cases where the transfer time is not negligible apart from the processing time. The paper reviews whether we can describe the implemented computing processes by using the accurate interpretation of the computing model, and whether we can explain the issues experienced in different fields of today's computing by omitting the wrong omissions. Furthermore, it discusses some of the consequences of improper technological implementations, from shared media to parallelized operation, suggesting ideas on how computing performance could be improved to meet the growing societal demands.

**Keywords:** generalized computing paradigm; dispersion; computing efficiency; temporal logic; time-aware computing; neuronal computing





## 1. Introduction

Today's computing science commonly refers to the classic Electronic Discrete Variable Computer [1] (EDVAC) report "First Draft" [2] as its solid base. von Neumann performed a general analysis of computing; however, for the intended vacuum tube implementation only, he made some very strong omissions. He limited the validity of his "*procedure*" (but not of his model!) for vacuum tubes only, carefully drawing the range of validity of the "initial approximations" he used. He made clear that using "too fast" vacuum tubes (or other elementary processing switches) *vitiates* his simplified model. Furthermore, he emphasized that it would be *unsound* to apply his simplified model to modeling a neural operation, where the timing relations are entirely different. Unfortunately, sometimes an implementation based on a different physical effect or material is called a 'model'; for examples, see [3]. Similarly, changing parts of the architectural principles are called models (see cited references in [4] or [5]). We use the word for the *abstract description of computing*, and discuss some of its different implementations in the paper.

Actually, his statement was that

```
if [timing relations of] vacuum_tubes
then Classic_Paradigm;
else Unsound;
```

In that phase of development, and in the age of vacuum tube technology, he did not feel the need to work out the "procedure" that we could follow when the processing elements get faster, and the timing relations do not enable us anymore to neglect transmission time apart from processing time. However, he strongly emphasized that we must revise the computing paradigm (especially the omissions he made about the timing relations)

according to technological development, whether the conditions of applying the "classic paradigm" are still satisfied.

The technology, with the advent of transistors and integrated circuits, quickly forgot vacuum tubes. The stealthy nature of the impressive technological development covered for decades was that the computing paradigm, created with vacuum tube timing relations in mind, was not valid for the new technology. The experts early noticed that the development of computing had slowed down dramatically [6]. Many experts suspected that the computing paradigm itself, "*the implicit hardware/software contract* [7]", was responsible for the experienced issues: "*No current programming model is able to cope with this development [of processors], though, as they essentially still follow the classical van Neumann model*" [8]. However, when thinking about "advances beyond 2020", on the one hand, the solution was expected from the "*more efficient implementation of the von Neumann architecture*" [9]. On the other hand, it was stated that "*The von Neumann architecture is fundamentally inefficient and non-scalable for representing massively interconnected neural networks*" [10].

The operating, technological, and utilization characteristics of computers have drastically changed. The aspects of utilization, developing architecture, using different technologies, and new physical effects/materials have been extensively discussed (see, for example, cited references in [3]). However, a new theoretical basis for the detailed analysis of the effects of those changes is still missing; those aspects have been discussed using the classic computing paradigm, which is definitely 'unsound' to describe them.

The methodology used here is to discuss the 'first principles' of the computing process in an implementation-independent way. That is, the computing process is considered as an aligned and constrained sequence of transfer and processing phases; logical computability is provided independently from the alignment (in a time-unaware way, such as the different Turing machines) or a physical/biological event (delivered by a material carrier). We illustrate the theoretical discussion with case studies taken from different implementation technologies.

## 2. von Neumann's Ideas

### 2.1. The "von Neumann Architecture"

The great idea of von Neumann was that he defined an interface (an abstract model) between the physical device and its mathematical exploitation [2]. His publication is commonly referred to as "von Neumann architecture" (perhaps his report is mismatched with the other report on Electronic Discrete Variable Computer [1] (EDVAC) [1]). However, in the first sentence, he makes it clear that "*The considerations which follow deal with the* **structure** *of a very high speed automatic digital computing system, and in particular with its* **logical control**." He did *not* define any architecture; just the opposite: he wanted to describe the operation of the engineering implementation in abstract terms, enabling mathematics to establish the science of computing. Moreover, "this is . . . the first substantial work . . . that clearly *separated logic design from implementation.* . . . The computer defined in the 'First Draft' *was never built, and its architecture and design seem now to be forgotten.*" [11].

### 2.2. The Model of Computing

"*Devising a machine that can process information faster than humans has been a driving force in computing for decades*" [4]. The model of computing did not change during the times, for centuries:

- The input operand(s) need to be delivered to the processing element;
- The processing must be completely performed;
- The output operand(s) must be delivered to their destination.

The processing elements must be notified that all of their needed operands are *available*, and the result can be delivered to its destination only if the operation is *completed*. Whether it is biological, neuromorphic, quantum, digital, or analog, the processing cannot even begin before its required input operands are delivered to the place of operation, and vice versa; the output operand cannot be delivered until the computing process terminates.

*Phases of computing logically depend on each other, and in their technical implementations, they need a proper alignment: synchronization.* In this way, *the data transfer and data processing phases block each other.* The effective computing time of the model comprises both the transfer time(s) and the processing time. Even if, in the actual case, one of them can be neglected apart from the other.

In most cases, simple operations are chained (even if the same physical processing unit performs them), and several processing units must cooperate. The chained processing units receive their input only when the module they receive their input from finishes its operation. That is, *any technological implementation converts the logical dependence of their operations to temporally consecutive phases: the signal of one's output must reach the other's input before the chained computation can continue.*

This limitation is valid for all implementations, from geared wheels to transporting electrons/ions or changing quantum states. An electronic computer is no exception, although its operation is too fast to perceive with our human sensors. When designing computing accelerators, feedback, and recurrent circuits, this point deserves special attention: the computation considers the corresponding logical dependence through its timing relations.

According to the computing model (see Figures 1 and 2), we assume that: we have the prepared operand(s) (data) in the input section of the computing unit (we assume that the operands are directly available), the data processing needs time, delivering the result to its output section needs time, and the operations and events are properly aligned. Correspondingly, the signal "Begin computing" is provided when the needed operands are available, and on terminating the operation, an "End computing" signal is provided. Between these two signals, the computing unit is busy: a "Computing cycle" is in course. Notice that the result is not necessarily available yet in its output section. Notice that, here, we tacitly assume that the processing unit is always available when needed; a consequence of the Single Processor Approach (SPA) [12]. Given that the timing of the "Operand available" signal varies with the access mode (for main memory and even for cache memory, it is longer than the "Computing cycle"), several computing chains (threads) can share the computing unit. Introducing hardware (HW)-threads (or hyper-threads) only increases the utilization of the computing process [13], but does not need another computing model. The diagram lines are somewhat similar to an electronic timing diagram, but instead, they represent a proper alignment of processes.

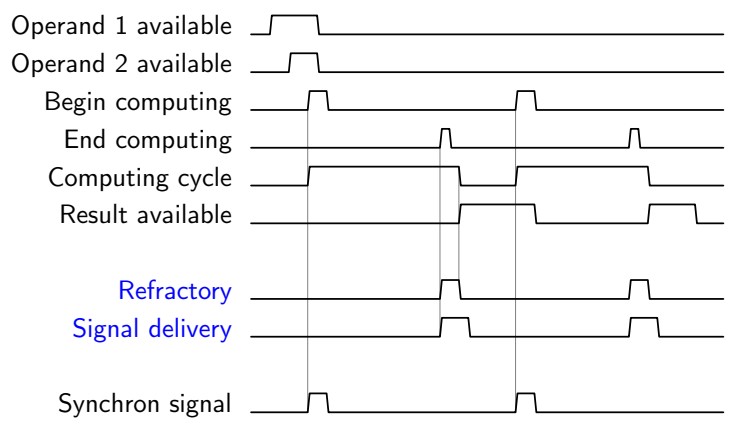

**Figure 1.** Timing relations of von Neumann's simplified (incomplete) timing model: the data transfer time neglected apart from the data processing time; synchronization can have small dispersion.

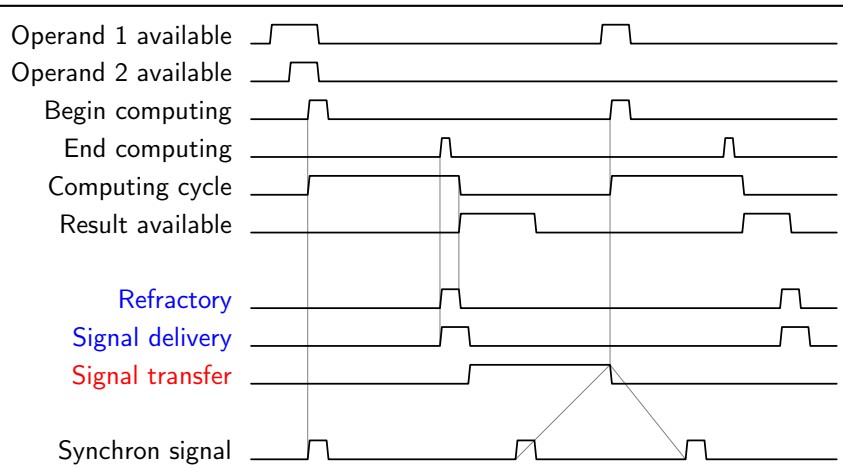

**Figure 2.** Timing relations of von Neumann's complete timing model, with data transfer time in chained operations; synchronization becomes an issue as the physical size of the computing system grows.

### 2.3. The Computer and the Brain

von Neumann discussed the operation of neurons in parallel with the operation of his intended technological implementation. In this context, it is clear that his model was *biology-inspired*, but (as can be concluded from the next section) because of its timing relations, it was not *biology-mimicking*. It is just a subtlety of the technological development that today's electronic computing devices work using timing relations more similar to our brains than vacuum tubes (and the mathematical abstraction based on it). Of course, there exists a resemblance in their operations, but handling temporal behavior also manifests in crucial differences [14]; among others—that our brain works asynchronously and auto-synchronized, while our computing systems work (mostly) in a synchronized way. These differences also explain why biology can store information natively (without needing a special memory unit), why biological systems can natively learn, and why technological systems cannot [15]; furthermore, why using technological systems for understanding learning and behavior in biological systems are mostly irrelevant [16].

### 2.4. Timing Relations

In his famous publication [2], von Neumann made a careful feasibility analysis and warned: "*6.3 At this point, the following observation is necessary. In the human nervous system, the conduction times [transmission times] along the lines (axons) can be longer than the synaptic delays [processing times]. Hence our above procedure of neglecting them aside from τ [the processing time] would be* ***unsound***."

von Neumann was aware of the facts and the technical development: "*we will base our considerations on a hypothetical element, which functions essentially like a vacuum tube. ... We reemphasize: This situation is only temporary, only a transient standpoint ... After the conclusions of the preliminary discussion, the elements will have to be reconsidered*" [17].

### 2.5. The Synchronous Operating Mode

As von Neumann explicitly warned in his "First Draft" [2] (Section 5.4), the operations must be synchronized appropriately. That is, computing faces further limitations inside its technical implementation. The computation operation phases must be appropriately synchronized; furthermore, the parallelization must be carried out with care. We can add: as well as the acceleration of computations, including computing feedback and recurrent relations.

The synchronization can be achieved by different means. The operand's availability must be signaled, anyhow: either on a per-operand basis (asynchronous operation) or using some central control unit (synchronous operation).

*2.6. Dispersion of the Synchronization*

In the same [2] (Section 5.4) von Neumann told "*We propose to use the delays τ as absolute units of time which can be relied upon to synchronize the functions of various parts of the device. The advantages of such an arrangement are immediately plausible*". When choosing such an absolute time unit, a "worst-case" timing must be selected, which *inherently introduces performance loss* for the "less wrong" cases. Technical systems, following von Neumann's proposal *for vacuum tube technology only*, typically use a central clock, and it is the designer's responsibility to choose a reasonable (but arbitrary) cycle period for the central clock. Figure 1 shows the timing relations assumed in the simplified (time-unaware) model. Notice that the synchronous signal mostly replaces the logical signals needed for the operation, which deserves special attention when designing computing accelerators, especially for processing periodical signals.

Figure 2 shows the timing relations of the complete (time-aware) model from Section 2.2. Notice the difference, where in chained computing operations, *the total computing times instead of the processing times* must be as uniform as possible. If either their *processing times* (complexity) or *transfer times* (connection technology, including different package handling, the signal's physical propagation speed, or the physical distance the signal has to pass) differ, the *total computing time* changes. In Section 4.1, we scrutinize the relations between the different timing contributions and their effects on the system's resulting performance.

## 3. Scrutinizing Dispersion

*The central synchronization inherently introduces some performance loss*: the processing elements will be idle until the next clock pulse arrives. The effect, of course, grows as the system's physical size grows or the processing time decreases apart from transfer time. This difference in the arrival times is why von Neumann emphasized: "*The emphasis is on the exclusion of a dispersion*" [2]. His statement in the previous section is true for the well-defined *dispersionless synaptic delay τ* he assumed, but not at all for today's processors, and even less for physically larger computing systems. The recent activity to consider asynchronous operating modes [4,18–21] is motivated by admitting that *the present synchronized operating mode is disadvantageous in the non-dispersionless world*.

von Neumann used the word "dispersion" only in a broad and mathematical sense, but he did not analyze its dependence on the actual physical conditions. Given that "*The emphasis is on the exclusion of a dispersion*" [2], we define a merit for the dispersion using the technical data of the given implementation. We provide a "best-case" and "worst-case" estimated value for the transfer time and define the dispersion as their geometric mean divided by the processing time.

*3.1. The Case of EDVAC*

von Neumann mentioned that a "too fast" processor—with his words—*vitiates* his paradigm. If we consider a 300 m² sized computer room and the 3000 vacuum tubes estimated, von Neumann considered a distance between the vacuum tubes, about 30 cm, as a critical value. At this distance, the transfer time is more than three orders of magnitude lower than the processing time (neighboring vacuum tubes are assumed). The worst case is to transfer the signal to the other end of the computer room. With our definition, the dispersion of EDVAC is (at or below) 1%.

*3.2. The Case of Integrated Circuits*

We derive dispersion for integrated circuits in the same way as discussed for vacuum tubes. Figure 3 shows the dependence of different dispersion values on the year of fabrication of the processor. The technical data are calculated from publicly available data (https://en.wikipedia.org/wiki/Transistor_count, accessed on 7 July 2021) and from [1]. The figures of the merits are rough and somewhat arbitrary approximations.

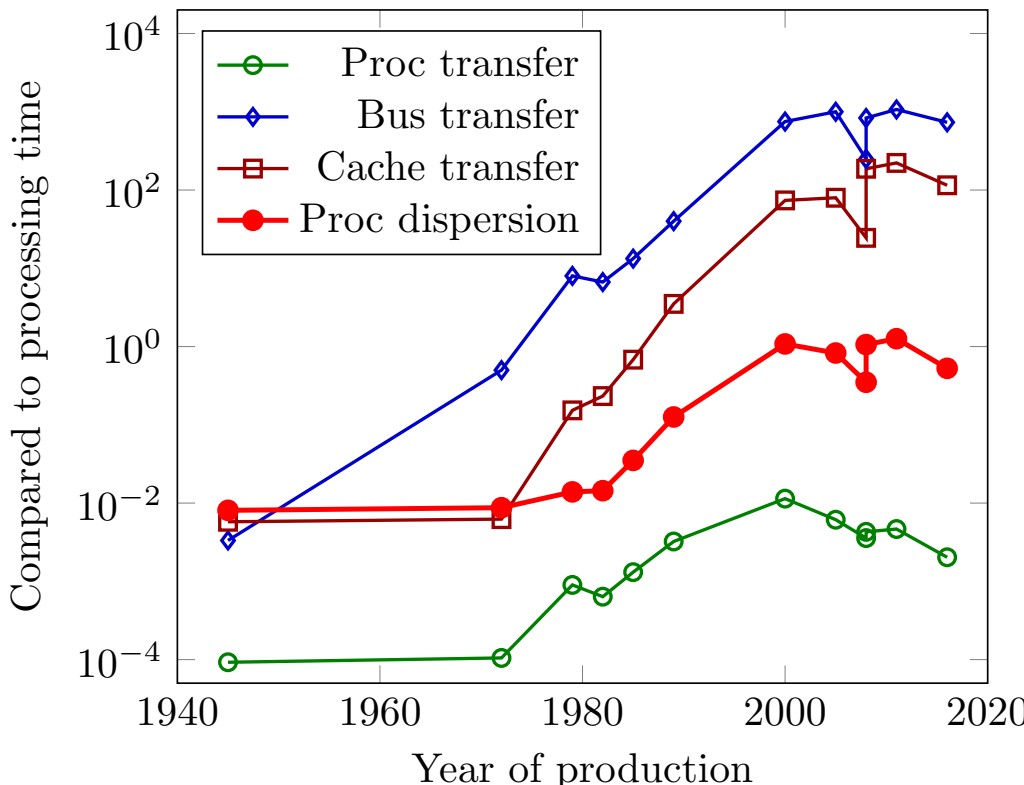

**Figure 3.** The history of some relative temporal characteristics of processors, in function of their production year. Notice how cramming more transistors in a processor changed their temporal characteristics disadvantageously.

We estimate the distance between the processing elements in two different ways. We calculate the "average distance" of the transistors (the "best case") as the square root of the processor area divided by the number of transistors. We consider that, as a minimum distance, the signals must travel between transistors. Notice that this transfer time also shows a drastic increase with the number of transistors, but alone, it does not vitiate the classic paradigm. This value, divided by the distance the signal passes in a clock period, is depicted as "Proc transfer" in Figure 3. The maximum distance between the two farthest processing elements on the chip, the processor area's square root. Evidently, introducing clock domains and multi-core processors shades the picture. However, we cannot provide a more accurate estimation without proprietary technological data.

From the two distances, we derived the "Proc dispersion". As its diagram line shows, *in today's technology, the dispersion is near to unity.* That is, *we cannot apply the "dispersionless" classic paradigm anymore.* Reaching the plateau of the diagram lines coincides with introducing the "explicitly parallel instruction set computer" [22]: that was the maximum the classic paradigm enabled. Unfortunately, processors must use a (dozens of centimeters long) bus for communication with each other and their memory (see "Bus transfer"). At the point where the "Bus transfer" significantly increased compared to "Proc dispersion", cache memories appeared, and as the diagram line "Cache transfer" shows, decreased, by one order of magnitude, the system's effective dispersion.

As we have experienced, since the dispersion approached unity about two decades ago, only a tiny fragment of the input power can be used for computation; the rest of it is dissipated (produces only heat). *The dispersion of synchronizing the computing operations* vastly increases the cycle time, decreases the utilization of all computing units, and enormously increases the power consumption of computing [23,24]. It is one of the primary reasons for the inefficiency [25] of the ASIC circuits, and leads to the symptom that moving data requires more energy [26] than manipulating it. The increased *dispersion* enormously *decreases performance* as the complexity and the *relative transfer-to-processing time* increases.

### 3.3. The Case of Technology Blocks

Given that processing elements and storage elements are usually fabricated as separated technological blocks (this technical solution is misinterpreted as "the von Neumann architecture": the *structure* and *architecture* are mismatched), the blocks are connected by wires (aka bus), we also estimated a "bus transfer" time. The memory access time in this way is extended by the bus transfer time. We assumed that cache memory is positioned at an average distance of half processor size because of this effect. This time is shown as "Cache transfer" time. The cache memories appeared at about the end of the 1980s, when it became evident that the bus transfer drastically increased the memory transfer time (our virtual cache timing can be calculated for all processors, however).

An interesting parallel is that both EDVAC and Intel 8008 had the same number of processing elements. The relative derived processor and cache transfer times are in the same order of magnitude. However, notice that the bus transfer time's importance had grown and started to dominate personal computers' single-processor performances. A decade later, the physical size of the bus necessitated introducing cache memories. The physical size led to saturation in all relative transfer times. The real cause of the "end of the Moore age" is that *Moore's observation is not valid for the bus's physical size*. The slight decrease in the relative times in the past years can probably be attributed to the sensitivity of our calculation method to the spread of multi-cores; this suggests repeating our analysis method with proprietary technological data.

### 3.4. The Need for Communication

Moreover, at the time when von Neumann proposed his paradigm, there was only one processor. Today, billions of processors are fabricated every year. They use ad-hoc methods and ideas about *cooperation and communication*; furthermore, they use their *payload times* for that activity. That activity increases the non-parallelizable portion of the tasks. It is very desirable to extend the computing paradigm with considering the presence of other processors.

### 3.5. Using New Physical Effect/Technology/Material in the Computing Chain

Given that the effective computing time also comprises the transfer time, the meaningful analysis must consider the full-time budget of the computing operations.

#### 3.5.1. Quantum Computing

von Neumann, of course, could not foresee the dawn of modern electronic technology. His discussion, by intention, did not consider the already known quantum logic [27] with its probabilistic outcomes. However, today, researching quantum computing has become very intensive. Its principle enables solving only particular problems, rather than building general-purpose computers. They run "toy algorithms" with accuracy 50–90%, and "the main challenge in quantum computing remains to find a technology that will scale up to thousands of qubits" [28]. Quantum computing has shown increasing advances in the past few years. Still, since the number of qubits needed to perform its targeted operation is estimated as 100,000, the technical difficulties make it elusive to put together a sufficiently large number of qubits.

In its most relevant field "the current state of quantum computing and progress is highly unlikely to be able to attack RSA 2048 within the next decade" [29]. The algorithmic improvements restrict their usability inside that field; furthermore, they cannot form the basis for general-purpose computers. It is possible to imitate von Neumann circuits [30] using qubits, but they inherit their limitations as well. It is also possible to imitate classical computing circuits using quantum Fourier transform [31]. Still, the synchronization issues (both 'Begin computing' and 'End Computing', as well as 'Operand available' signals) remain out of scope.

As discussed in [32,33], for the adequate description of computing, three-state systems must be used. In the present two-state digital electronic logic systems, discharging the in-

ternal capacitances can be considered a "refractory" period, i.e., a third state, which defines time's direction. However, it is not known if such a third state can be available among the quantum states at all. Because of these reasons, in the foreseeable future, quantum computers will not represent an alternative general-purpose architecture. "Building such machines are decades away" [28].

### 3.5.2. Biomorphic Architectures

Given that, in his classic report, he outlined that his proposal was about *the logical structure* of a computing implementation, it is hard to interpret what the "von Neumann architecture" means if one speaks about non-von Neumann architecture in connection with biomorphic computing (for a review, see [4]) or emerging non-Neumann architectures [5]. In regard to the *need for manufacturing technology blocks*, (such as Central Processing Unit (CPU) and memory), it is preferable to use an architecture (the "von Neumann architecture"). Still, it has as little to do with the *computing model* as it is the need for *wiring those blocks technologically* with a single shared bus (the "von Neumann bottleneck").

The third state is also missing here; so these architectures may inherit conventional system inability to simulate the three-state system with two-state elements. In addition, those systems also inherit some technology solutions, as discussed below. This is why judges of the Gordon Bell Prize noticed that *"surprisingly, [among the winners of the supercomputer competition] there have been no brain-inspired massively parallel specialized computers"* [34].

### 3.5.3. Artificial Neural Networks

From the computational point of view, an Artificial Neural Network (ANN) is an adaptive distributed processor-based architecture widely used to utilize inputs and simulate human-processing, in terms of computation, response, and decision-making. Their operating principle undergoes the general distributed processing principles. As discussed in [35], they can do valuable work at a small number of cores ('toy level') and can be useful embedded components in a general-purpose processor, but have severe performance limitations at large scale systems. They are sensitive to the synchronization issues discussed here, primarily if they use feedback and recurrency [16].

### 3.5.4. Using Memristors for Processing

Recently proposed ideas involve replacing slow digital processing with quick analog processing [36–38], to consider any future new physical effects and/or materials [3]. They decrease the physical processing time, *only*. To make them useful for computing, both the in-component transmission time, and especially the inter-component transmission time, must be considerably decreased.

It sounds good that "*The analog memristor array is effectively the neural network laid out in the form of a crossbar, which can perform the entire operation **in one clock cycle**"* [21]. In brackets, however, fairly added that *"(not counting the clock cycles that may be required to fetch and store the input and output data)"*. All operands of the memristor array must be transferred to its input section (and previously, they must be computed or otherwise produced); furthermore, the results must be transferred from their output sections to their destinations. This requirement persists even if continuous-time data representation [38] is used, and may require hundreds of additional clock cycles. One shall compare the memristor-related operations' effective computing time to conventional operations' effective time, from the beginning to the end of the computing operation, to make a fair comparison.

One can easily misidentify the temporal behavior of components and their material in the time-unaware model. Five decades ago, even *memristance* was introduced [39] as a fundamental electrical component, meaning that the memristor's electrical resistance is not constant, but depends on the history of current that had previously flowed through the device. There are, however, some serious doubts as to whether a genuine memristor can actually exist in physical reality [40]. In light of our analysis, some temporal behavior exists;

the question is how much is related to material or biological features, if our time-aware computing method is followed.

### 3.5.5. Half-Length Operands vs. Double-Length Ones

The mutual blocking of the transfer (and other, non-immediately payload operations) and the payload operations similarly lead to disappointing efficiency improvement when one attempts to use half-length operands instead of double-length ones. The expectation behind the idea is that the shorter operand length may increase by a factor of four the desperately low efficiency of the artificial intelligence class applications running on supercomputers. One expects a four-fold performance increase when using half-precision rather than double-precision operands [41], and the power consumption data underpin that expectation. However, the measured increase in computing performance was only slightly more than three times higher: its temporal behavior limits the utility of using shorter operands, too. The housekeeping (such as fetching, addressing, incrementing, branching) remained the same, and because of the mutually blocking nature of the payload-to-non-payload operations, the increase of the payload performance is significantly lower. In the case of AI-type workload, the performance with half-precision and double precision operands differ only marginally for vast systems. For details, see [35,42].

### 3.5.6. The Role of Transfer Time

The relative weight of the data transfer time has grown tremendously for many reasons. Firstly, because of miniaturizing processors to sub-micron size, while keeping the rest of the components (such as buses) above the centimeter scale. Secondly, the single-processor performance has stalled [43], mainly because of reaching the limits, the laws of nature enable [44] (but, as we present, also because of tremendously extending its inherent idle waiting times). Thirdly, making truly parallel computers failed [7], and replaced with *parallelized sequential computing* (aka distributed computing), disregarding that the operating rules of the latter [42,45] sharply differ from those experienced with segregated processors. Fourthly, the utilization mode (mainly multitasking) forced us to use an operating system (OS), which imitates a "new processor" for a new task, at serious time expenses [46,47]. Finally, the idea of "real-time connected everything" introduced geographically large distances with their corresponding several millisecond data transfer times, while "Big Data" assumes that everything is in cache memory. Despite all of this, the idea of non-temporal behavior was confirmed by accepting the concept of "weak scaling" [48], suggesting that *all housekeeping times, such as organizing the joint work of parallelized serial processors, sharing resources, using exceptions, and OS services, delivering data between processing units and data storage units, are negligible*. See [35] why weak scaling is wrong. Essentially, this is why the algorithmic scalability assumes a dependence on the number of operations (i.e., it assumes that the transfer time can be neglected aside from processing time), rather than taking into account how the *effective computing time* changes with the *transfer time* between the computing units as the physical size of the system increases.

### 3.5.7. How the Presence of Transfer Time Was Covered

The experience showed that wiring (and its related transfer time) has an increasing weight [44] in the timing budget even within the core. When reaching the technology limit of about 200 nm around the year 2000 (https://en.wikipedia.org/wiki/Transistor_count), accessed on 7 July 2021, wiring started to dominate [49] (compare this date to the year when saturation was reached in Figure 3). Further miniaturization can enhance computing performance only marginally, but increases the issues due to approaching the limiting interaction speed, as discussed below.

*"To compensate for the different number of gates on different paths, functionally not needed (such as invert–reinvert) gates are inserted on the path with fewer gates. The design comprises several clock domains, and some more extensive parts of the design run with different synchronization. However, approximately 30% of the total consumption of a modern microprocessor is solely used*

*for the clock signal distribution*" [23] (and about the same amount of power is needed for cooling). Furthermore, the difference in the length of physical signal paths causes a "skew" (*dispersion*) of the signals, which become significant challenges in designing high-performance systems [50]). Even inside the die: the segregated processors have very low efficiency [25]. Despite this, today, wafer (and even multi-wafer-sized [51]) systems are also under design and in use.

In complete systems, such as supercomputers running High Performance Conjugate Gradients (HPCG) workloads, only 0.3 % of the consumed energy goes for computing and this proportion will get much worse if conventional systems attempt to mimic biology, such as running an ANN workload. The poor understanding of basic terms of computing resulted in that in supercomputing "*the top 10 systems are unchanged from the previous list*" [52], and that "*Core progress in Artificial Intelligence (AI) has stalled in some quite different fields*" [53]; from brain simulation [54] to ANNs [55]; in general, the AI progress [53] as a whole. Considering temporal behavior is a must [56].

## 4. The Time–Space System

von Neumann—at his time, and in the age of vacuum tube technology—did not feel the need to discuss what a procedure can justify, describing the computing operation in a non-dispersionless case. However, *he suggested reconsidering the validity of the neglections he used in his paradigm for any new future technology*. The real question [50] is, the discussion of which is *missing from the "First draft", what procedure shall we follow if the transfer time is not negligible*?

### 4.1. Considering the Transfer Time

Although he explicitly mentioned that the propagation speed of electromagnetic waves limits the operating speed of the electronic components—until recently—that effect was not admitted in computing (except introducing clock domains and clock signal skew). In contrast, in biology, the "spatiotemporal" behavior [57] was recognized very early. In both technical and biology-related computing, the recent trend has been to describe computing systems theoretically and model their operations electronically using the time-unaware computing paradigm proposed by von Neumann, which is undoubtedly not valid for today's technologies. Furthermore, as mentioned above, *our computing devices' operating regimes are closer to our brains than the abstract model*. That is, *a similar description for both the computer and the brain would be adequate*.

Fortunately, the spatiotemporal behavior suggests a "procedure" that can be followed in the case when the transfer time can even be longer than the processing time; a point that is missing from the "First Draft" [50]. Although biology—despite the name "spatiotemporal"—describes the behavior of its systems using separated space and time functions (and as a consequence, needs ad hoc suggestions and solutions for different problems), it has one common attribute with technical computing: in both of them, the information transfer speed is limited (although several million times lower in biology). For its physical (and maybe philosophical) relevance, a more detailed explanation see in [58]. Giving an explicit role to time also reveals why information storing and learning have a quite different implementation and behavior in biological and technological neural computing [14]. Furthermore, it explains that methods of learning and machine learning are orthogonal [16], and so are intelligence and machine intelligence.

To introduce time into computing in explicit form, we can use a 4D coordinate system (for a mathematical exposition see [59]), where, in addition to the three spatial coordinates, we use the time as the fourth coordinate. Such geometry is known in physics as Minkowski geometry. For the first look, it seems strange to describe such systems with Minkowski coordinates, given that it became famous in connection with Einstein's theory of special relativity. However, in its original form, only the existence of a limiting speed is assumed, and modern treatments of special relativity base it on the single postulate of Minkowski spacetime [59].

The 4D coordinate system enables us to describe the correct behavior of information processing in science-based technological implementations and biology for any combination (ratio) of the transfer time and the processing time. The key idea is similar to that of the Minkowski coordinates, but we use the scale factor differently. Changing the way of using a scale factor (the interaction speed) does not change the signature of the geometry, so our proposed method is equivalent to the technique introduced by Minkowski. The difference is that we transform the spatial distances between computing components (which can be *Si* gates, cores, network nodes, biological or artificial neurons) to temporal distance (measured with the limiting speed along the signal path) instead of transforming the time coordinate to space coordinate. On different abstraction levels, the "computing component" is defined differently, so we formulate the terms with care.

We only assume that a limiting speed exists, and that *given that it has a material carrier, transferring information in the system needs time*. In our approach, *Minkowski provided a mathematical method to describe information transfer phenomena in a world where the interaction speed is limited*. The only new assumptions we make are that the events also have processing times, such as an atomic transition, executing a machine instruction, or issuing/receiving a neural spike; furthermore, the interaction speed may be other than the speed of light. Special relativity describes the space around us with *four-dimensional space–time* coordinates, and *calculates the fourth spatial coordinate from the time as the distance the light traverses in a given time*; simply because, around us, the distance was the easily accessible, measurable quantity—one hundred years ago.

In computing, distances get defined during the fabrication of components and assembling of the system. They may be different in different designs; however, they must meet their *timing constraints*. In biological systems, nature defines neuronal locations and distances, and in 'wet' neurobiology, *signal timing rather than axon length is the right (measurable) parameter*. To introduce *temporal logic* (meaning: the logical value of an expression depends on WHERE and WHEN it is evaluated) into computing, a different approach to the 4D coordinates is required: the time, rather than the position, is the primary measurable quantity. We need to use a special four-vector, where *all coordinates are time values*: the first three are the corresponding local coordinates (divided by the speed of interaction), having a time dimension. The fourth coordinate is time itself. *Distances from an event's location are measured along their access path; they are not calculated from their corresponding spatial coordinates*.

In electronics, the limiting speed is connected to the speed of light (in biological systems, the issue is more complicated, but the speed of interaction is finite, anyhow). Given that we use the *time* as the primary variable, we can use the formalism to describe neuronal operation (where the conduction velocity is modulated) with time rather than position. However, in the latter case, the formalism is less straightforward (and enables us to understand how information is stored and learning occurs in biology [14]). To illustrate the effect of introducing transfer time into computing, we describe a thought computing experiment resembling Einstein's example of a moving observer in physics. However, we have only one frame of reference, and our observers (the computing objects) are not moving, only the information carrier they send to each other. With that thought computing experiment, we demonstrate that the solid mathematical background connected to the Minkowski coordinates (and all associated behavior of modern science, also describing modern technological materials) is preserved in our slightly different coordinate representation. Our *time–space* system is equivalent to the commonly used *space–time* systems. In the following sections, we use the transformed coordinates only to describe computing systems' temporal behavior.

*The only change introduced to logic functions of computing is that they are not any more evaluated implicitly at point (0,0,0,0) (In other words, the classic paradigm is valid only for infinitely small and infinitely fast technical computers). Instead, they are evaluated at a point (x,y,z,t) of the time–space*. Below, we introduce the idea and its notations. The validity and the mathematical features of the space–time systems have been scrutinized exclusively

in the past 120 years. So, on the one hand, we use the solid background the computer science is based upon: mathematics; on the other hand, we extend it with the similarly solid background of *time–space* formalism.

### 4.2. Introducing the Time–Space System

With the reasoning above, we introduce [58] a *four dimensional time–space* system. The resemblance with the Minkowski space is evident, and the name difference signals the different utilization methods.

In our particular coordinate system (used in some figures below), formally (x,y,t) coordinates are used (for better visibility, the third spatial coordinate is omitted). What happens at time *t* in a computing component at position (x,y) occurs along a line parallel with axis *t*, along a line perpendicular to plane at (x,y). *The objects are annotated with their spatial position coordinates 'x' and 'y', but they are time values: how much time the signal having the limiting speed needs to reach that point.* Several 'snapshots' taken at time *t* about the same computing object are shown in the 3-D figures on top of each other. The computing objects may alternatively be positioned at some arbitrary position that corresponds to the same *time distance* from point (0,0,0) (a cylindrical coordinate system would be adequate but would make both visualization and calculations much harder to follow).

The arrows in the same horizontal plane represent the same time (no transmission). *The interaction vectors are neither parallel with the time axis nor are in a spatial plane: both their temporal and spatial coordinates change as the interaction propagates. In electronic systems, the speed of interaction is constant, so the vectors are simple lines. Those spatial vectors are displayed with their different projections in the corresponding figures, enabling their easy trigonometric calculation.* The horizontal blue arrows are just helper lines: the position (annotated by x,y, but denoting the time the signal from (0,0,0) needs to reach this position) is projected to *time* axis and the XY plane. The red arrow points from the coordinates of the beginning to the coordinates of the end of an action. It is a Minkowski distance, and it has the dimension of time. In some cases, it may approximate the distance of events in the Minkowski space; it connects propagation and processing times. They are measured on different axes, and the basic trigonometry is valid here. The result of the operation is available at a position different from that of its operand. It needs additional time to reach an observer. The red arrow is the vectorial sum of the two projections, also in that plane.

At the positions in the (x,y) plane, the events happen at the same time. The other processing units will notice that event at a correspondingly later time in another XY plane. The events happening in connection with a processing unit are aligned on an arrow parallel with axis *t*. In this sense, *we can interpret 'classic computing': our objects are compressed to one XY plane: all events happen simultaneously at all objects. Even as the time distance between our computing objects is zero because of the instantaneous transmission (independently from its technical implementation), the mathematical point (0,0,0) will represent the figure.* The processing time is only an engineering imperfectness, according to the 'classic computing science'.

### 4.3. Validating the Time–Space System

Figure 4 represents essentially a light cone in 2D space plus a time dimension. On the one hand, it demonstrates that our procedure using 4D coordinates with a different scaling factor is appropriate for discussing temporal behavior in our thought experiment. On the other hand, it shows *why time must be considered explicitly in all kinds of computing*. The figure shows that an event appears in our *time–space system* at point (0,0,0). The only (very plausible) difference to the classic relativistic thought experiment is that we assume that signaling needs some time, too (not only the interaction but also its technical implementation needs time). Our observers (fellow computing objects) are located on the 'x' axis (unlike in the relativistic thought experiment, they are not moving); the vertical scale corresponds to time. In the thought computing experiment, light is switched on in origin: the need to perform a calculation appears. The observers switch their light on (start their calculation) when they notice that the first light is switched on (the instruction/operand reaches their

position). The distance traveled by the light (the signal carrying the instruction) is given as the value of time multiplied by the speed of light (signal speed). At any point in time on the vertical axis, a circle describes light propagation (signal). In our (pseudo) 3-dimensional system, the temporal behavior is described as a conical surface, known as the *future light cone* in science.

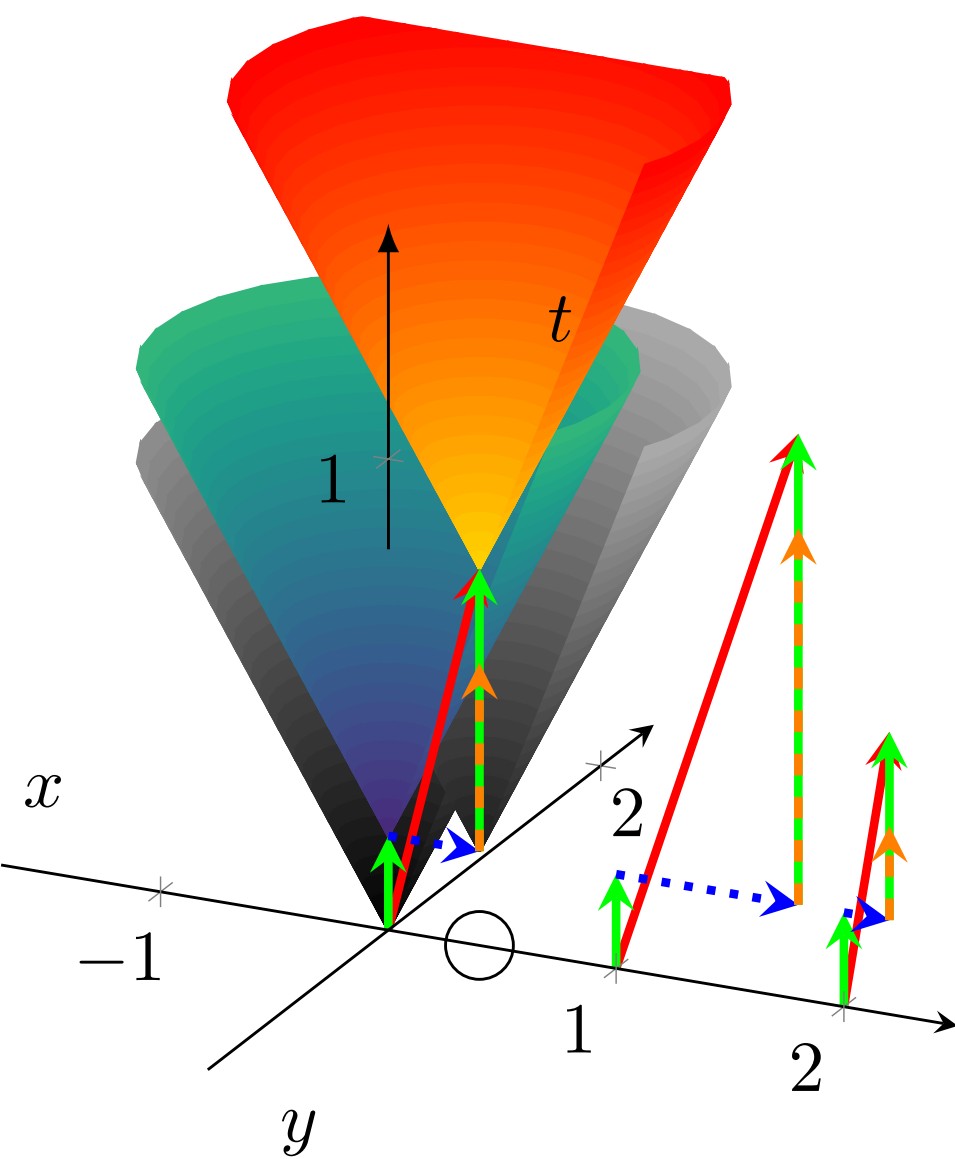

**Figure 4.** The computing operation in *time–space* approach. The processing operators can be gates, processors, neurons, or networked computers. The "idle waiting time", rooting in the finite interaction speed and the physical distance of computing elements (see mixed-color vectors in figure), is one of the major sources of computing systems' inefficiency.

Both light sources (computing objects) have some 'processing times', that passes between noticing the light (receiving an instruction/operand) and switching their light (performing an instruction). An instruction is received at the bottom of the green arrow. The light goes on at the head of the arrow (i.e., at the same location, but at a later time) when the 'processing time' $T_p$ passed. Following that, the light propagates in the two spatial dimensions as a circle around the axis 't'. Observers at a larger distance notice the light at a later time: a 'transmission time' $T_t$ is needed. If the 'processing time' of our first

event's light source were zero, the light would propagate along the gray surface at the point (0,0,0). However, because of the source's finite processing time, the signal propagates along the blueish cone surface at the head of the green arrow.

A circle marks the position of our other computing unit on the axis 'x'. With zero 'transmission time', a second gray conical surface (at the head of the horizontal blue dotted arrow) would describe the propagation of its signal. However, this second 'processing time' can only begin when our second processing unit receives the instruction at its position: when the mixed-color vertical dashed arrow hits the blueish surface. At that point begins the 'processing time' of our second processing unit; the yellowish conical surface, starting at the second vertical green arrow, describes the second signal propagation. The horizontal (blue dotted) arrow describes the second computing unit's physical distance (as a time coordinate) from the first one. The vertical (mixed color dashed) arrow represents the time delay of the instruction. It comprises two components: $T_t$ transmission time (mixed color) to the observer unit and its $T_p$ processing time (green). The light cone of the observer (emitting the result of calculation) starts at $t = 2 * T_p + T_t$. We use the red arrow to call attention to the effect: *the longer the red vector, the slower the system is*. The length of that vector might be of use as some summary of a computing system, especially for statistical purposes.

Two more observers are located on the axis 'x', at the same position, to illustrate the influence of transmission speed (and/or ratio *R*). For visibility, their timings are displayed at points '1' and '2', respectively. In their case, *the transmission speed differs by a factor of two* compared to that displayed at point '0'; in this way, three different $R = T_t/T_p$ ratios are used. Notice that at half transmission speed (the horizontal blue arrow is twice as long as the one at the origin) the red vector is considerably longer, while at double transmission speed, the decrease of the time is much less expressed. These insets illustrates the effect of transmission speed on observations. This phenomenon is discussed in detail in [42].

*4.4. Scrutinizing the Temporal Behavior*

The temporal behavior means that *values of logical functions depend on both time and place of evaluating the corresponding logical function*. Notice that some consequences stem immediately from the nature of our model. Now two computing elements are sitting at points (0,0,0) and (0,1,0). The second element calculates something that expects the calculation of the first element as its input operand of the second calculation. Consider that the result is inside the green arrow during its processing time and comes to the light after that time. As visible from the discussion and the figure, the first event happens at a well-determined position and time coordinates in our *time–space* system. Its spatial coordinates agree with those of the spatial coordinates of the first element. Furthermore, they are different from those of the second element. The result must be transported: it shall be delivered to the position of the second processing unit, which needs change in both position and time.

The event starts to propagate, and its final destination's coordinates differ in time and space from those of the origin. It would be described as positioned at the head of the red vector (such vectors are neither vertical vectors, parallel with the *t* axis, nor lie within the plane (x,y) ). During this transfer, the space coordinate (the projection of the *time–space* distance to the (x,y) plane) changes to the coordinates of the head of the blue arrow: here *was* the observer when the event happened. In the meantime, however, the time passed for the observer, and now the head of the vertical arrow at its position describes its coordinate. That vertical arrow comprises two contributions: the upper green arrow represents the processing time of the observer, and we also have the length of the mixed-color arrow (the idle waiting): it has the same length as the blue dotted arrow: our observer must wait for such a long time to receive the signal. Given that all signals are in the plane (x,t), the actual time distances can be calculated straightforwardly. The projection of the event to axis *t* is $T_p + T_i + T_p$. However, the position of the event is different from that of the beginning.

Hence, an observer receives the information later, depending on the position of the event and the interaction speed.

When looking at the events from the direction of the *x* axis in the (x,t) plane, we see that the total time corresponds to the sum of the two processing times (plus some unexplained idle time) in 'classic computing'. The waiting time of the second unit is only an arrowhead (instant interaction). It denotes the corresponding logical dependence but does not increase processing time. The second unit must wait for the result of the first calculation because of their logical dependence. However, because of the instant delivery, no additional waiting is required. Consequently, according to the "timeless (classical) paradigm", observers' distance seems to be zero. The introduced extra dimension, *time of interaction*, changes the picture drastically. The closer is the difference between the summed length of the two green vectors to the red vector's length, the more significant is their interaction speed. *They are equal, however, only if the interaction speed is infinitely large.*

### 4.5. Computing Efficiency as a Consequence of Temporal Behavior

It has been discussed—virtually infinite times in computing—that computing performance depends on many factors. However, the temporal dependence has never been included. As the discussion above suggested, the time has a decisive role in computing; the *idle waiting times* indeed degrade performance, so the rest of the discussion shall focus on the role of time in shaping performance.

Given that the spatial distance is equivalent to time, they have a similar role. The processor operation itself has some idle time, and, as discussed in the following sections, the outdated technical principles and solutions add more (and slightly different) idle times. Moore's observation is valid only for the gate's density inside chips, but not for their internal wiring and even less for the external wiring (such as buses) connecting them, so the idle (transmission) time increases. *This relation has a self-exciting effect: the low efficiency (that decreases as the required performance increases) means that a more significant portion of energy consumption is used for heating (and because of that: needs more cooling), needing more cooling that increases the physical distance of the components, causing worse performance, and so on.*

On the one hand, the communication between processors is implemented in a way that increases the non-payload, sequential-only portion of the task. On the other hand, the physical time of transmission (that depends on both the speed of interconnection and the physical distance between the corresponding computing objects) also significantly contributes to degrading the efficiency. Moreover, both the algorithms, how the components are used, and the architecture parameters all impact the system's computing efficiency. Systematic empirical investigations (for example [60]) found that for their specific application type "*the memory and the execution time required by the running are of $O(n^3)$ and $O(n^5)$ order*", in other words, *the empirical computing efficiency drops by two orders of magnitude*. Algorithm scaling is not possible without considering the temporal behavior of computing components.

*The modules' benchmark data define the hard limits, and their way of cooperation defines the soft limits we can experience.* That is, *the temporal behavior of their components is a vital feature of computing systems, especially of the excessive ones, mainly if they target high computing performance, especially if they are running very demanding workloads.* The effect of workload is why, especially for neural simulation, it was bitterly admitted that: "*artificial intelligence, ... it's the most disruptive workload from an I/O pattern perspective.*" (https://www.nextplatform.com/2019/10/30/cray-revamps-clusterstor-for-the-exascale-era/, accessed on 7 July 2021).

## 5. Technical Solutions for the Vacuum-Tube Age

Some of the "classic" technical implementations—due to the incremental development—survived their technological state-of-the-art, and (especially in the light of temporal analysis) need a drastic revision. Moreover, there is research (both in science and technology, with vast investments in the background) to find new materials/effects/technologies. However, science severely limits its usability: the temporal analysis provides a helping

hand and optimizes performance/cost. "Reinventing electronics" [20] is necessary not only for computing devices but also for their interconnection and modes of utilization.

### 5.1. Method of Identifying Bottlenecks of Computing

*The transmission time $T_t$ is an 'idle time'* from the point of view of computing: the component is ready to run, takes power, but does no valuable work. Due to their finite physical size and the finite interaction speed (both neglected in the classic computing paradigm), *the temporal operation of computing systems results inherently in an idle time of the Processing Units, which can be a significant contributor to the non-payload portion of their processing time. Given that the different effective processing times inevitably increase the dispersion, they can be a crucial factor of the experienced inefficiency of general-purpose chips [25].* With other significant contributors, originating from the technical implementation of their cooperation, these "idle waiting" times sharply decrease the payload performance of computing systems. It is worth discussing the inside-component and inter-component contributions separately.

In the spirit of the temporal behavior, we can set up two general classes of processing: the *payload processing $T_p$* makes our operations directly related to our goal of computation; all other processing is counted as *non-payload processing. The merit we use is the time spent with that processing*. As will be shown, some portion of the non-payload processing time is rooted in laws of nature: *computing inherently includes idle times*, some other part (such as housekeeping) is not *directly* useful. However, any processing takes time and consumes energy. The task of designing our computing systems is to reduce the effective processing time, i.e., *to develop solutions that minimize the proportion of the 'idle' activity; not only at the component level, but also at the system level.* Scrutinizing the temporal diagrams of components, solutions, and principles is an excellent tool to find bottlenecks.

In the figures below, near to the (vertical) axis *t* are shown vertical arrows (where payload processing happens) or lack of arrows (when non-payload processing occurs). The large amount of non-payload processing (that increases with the system's complexity) explains the experienced low computing efficiency of computing systems using those technical implementations. The proportions of times are chosen for better visibility and call attention to its effect rather than reflect some realistic arrangements.

### 5.2. Gate-Level Processing

Although for its end-users, the processor is the "atomic unit" of processing, principles of computing are valid also at a "sub-atomic" level, at the level of gate operations. (The reconfigurable computing, with its customized processors and non-processor-like processing units, does not change the landscape significantly.) Describing the temporal operation at gate level is an excellent example to demonstrate that *the line-by-line compiling (sequential programming, called also 'von Neumann-style programming' [61]), formally introduces only logical dependence, but through its technical implementation it implicitly and inherently introduces a temporal behavior, too.* The same is valid for any technical implementation of a Turing machine.

The one-bit adder is one of the simplest circuits used in computing. Its typical implementation comprises five logic gates, three input signals, and two output signals. Gates are logically connected internally: they provide input and output for each other. The relevant fraction of the equivalent source code is shown in Listing 1, while Figure 5 shows the timing diagram of a one-bit adder, implemented using common logic gates. This time, the computing objects are logical gates; one's output level serves as the input level for the other gate. The processing time is the time the gate needs to change its output level. The transfer times take their origin when the signal needs to propagate to another topological position along the wire. The coordinate system is the same as in Figure 4. The three input signals are aligned on axis *y*; the five logic gates are aligned on axis *x*. Gates are ready to operate, and signals are ready to be processed (at the head of the blue arrows). The logic gates have the same operating time (the length of green vectors); their access time distance includes the needed multiplexing. The signals must reach their destination

gate (dotted green arrows), which (after its operating time passes) produces the output signal, starting immediately towards the next gate. The vertical green arrows denote gate processing (one can project the arrow to axis $x$ to find out the gate's ID), labeled with the produced signal's name.

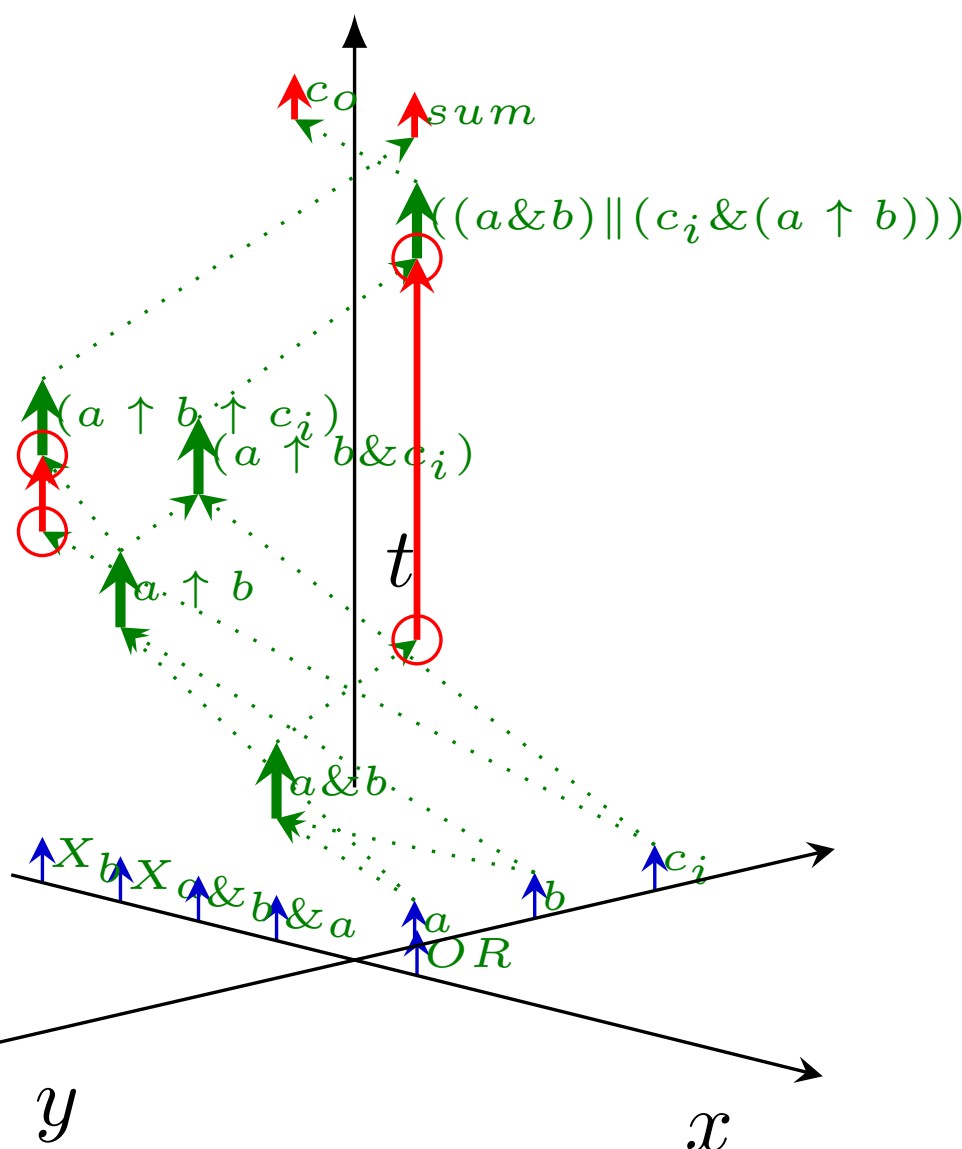

**Figure 5.** The temporal dependence diagram of a 1-bit adder. The diagram shows the logical equivalent of the SystemC source code of Listing 1, *the lack of vertical arrows* signals "idle waiting" time (undefined gate output), with "pointless" synchronization (red circles). The input signals $a$, $b$ and $c_i$ are aligned along axis $y$ (the input section), the computation takes place in gates aligned along axis $x$, and the output signals $c_o$ and *sum* are aligned again along axis $y$ (the output section).

Because of the non-synchronized operating mode, there are "pointless" arrows in the figure; see the red circles. For example, signal *a&b* reaches the *OR* gate much earlier than the signal to its other input. Depending on the operands of *OR*, it may or may not result in the final sum. The gates always have an output signal, even if they did not receive their expected input.

**Listing 1.** The essential lines of source code of the one-bit adder implemented in SystemC.

```
SC_MODULE (BIT_ADDER)
{
        sc_in<sc_logic> a,b,cin;
        sc_out<sc_logic> sum,cout;

        //Sensitivity to changes in a, b, & cin.
        SC_CTOR (BIT_ADDER)
        {
                SC_METHOD (process);
                sensitive << a << b << cin;
        }

        //Performed for any change
        void process()
        {
                //intermediate signals
                sc_logic aANDb,aXORb,cinANDaXORb;

                //Perform intermediate calculations
                //when any of the inputs changes
                aANDb = a.read() & b.read();
                aXORb = a.read() ^ b.read();
                cinANDaXORb = cin.read() & aXORb;

                //Calculate sum and carry out;
                // maybe not final ones
                sum = aXORb ^ cin.read();
                cout = aANDb | cinANDaXORb;
        }
};
```

Notice that according to Listing 1, "process()" is *sensitive* to all the three input signals. That is, every single time when any of the inputs changes, the "calculation" starts over and re-calculates all signal levels. Although only five switchings are needed for the operation, two more switchings happen, because of the undefined states: signals must pass a different number of gates. This number is only 1.4 times more than the minimum number needed for a single-bit adder. However, given that the adders provide result bits for each other, for a 64-bit adder, it is $1.4^{64}$, which is up to about 2 billion times more than the requested minimum (the actual number, of course, depends on the actual arguments). Given that the primary source of the energy consumption of gates is due to switching their states, *the actual power consumption of the 64-bit adder may be by orders of magnitude higher than that expected based on the operation of the time-unaware computing paradigm.* This effect may even more strongly contribute to wasting power for heating rather than computing than the clock distribution. Reducing the number of unneeded state transitions is vital for segregated single processors and large computing systems.

*5.3. Design Aspects*

The present conclusions shall be used in electronic design in two stages. In the first stage, one can analyze the existing designs, components, and principles, understand their inherited weaknesses (see following sections), and make them more effective.

In the second stage, one needs to consider the old truth that "More is different" [62]. We shall revise to apply the design principles developed for "toy level" systems to the present vast electronic systems. As the examples show, the technical implementation of the serial bus (even if it is a high-speed one); replacing parallel processing with serialized

sequential processing; proposing in-memory computing without establishing its component and architecture base, etc., represent performance limits for computing. As discussed above, increasing the bit width of an adder circuit increases its power consumption disproportionally. In addition to developing more low-energy circuits, it should be realized (as some technical implementations partly do) that the higher bit width needs different design principles.

The design should be more system-centric rather than working along cut and paste logic. The idea is somewhat similar to FPGAs Programmable I/O Cell (PIC): at individual simple circuits' level, their use seems to waste resources. However, at the system level, they are beneficial, although relatively expensive, resources. Somewhat closer to the subject, the present design uses clock domains to cover the experienced "skew" of clock signals, which largely contributes to wasting energy for heating. From the complexity of circuits, investing extra resources in auto-synchronization pays back in power consumption and operating speed.

Biology uses three-state logic, which enables shallow energy consumption. Neurons are "off" until their synapses receive some input, then for some time, they get "open", after some time "inactivated", and after some time "off" again. As discussed in [14], the three-state operation mode is desirable from several points of view. The present design point of view, replacing the "inactivated" state with a "down" edge, enables reaching higher operating frequency [63]. Still, large designs shed light on the limitations caused by attempting to replace the energetically needed three-state operation [32,33] with a simplified two-state operating mode.

*5.4. The Serial Bus*

Figure 6 discusses, in terms of "temporal logic": why using high-speed buses for connecting modern computer components leads to very severe performance loss, especially when one attempts to imitate neuromorphic operation. The processors are positioned at $(-0.3,0)$ and $(0.6,0)$. The bus is at position $(0,0.5)$. The two processors make their computation (green arrows at the place of processors), then they want to deliver their result to its destination. We assume that they want to communicate simultaneously. First, they must have access to the shared bus (red arrows). The core at $(-0.3,0)$ is closer to the bus, so its request is granted. As soon as the grant signal reaches the requesting core, the bus operation is initiated, and the data starts to travel to the bus. As soon as it arrives at the bus, the bus's high speed forwards it, and at that point, the bus request of the other core is granted, and finally, the computed result of the second core is bused.

At this point comes into the picture the role of workload on the system: we presumed that the two cores want to use the single shared bus, at the same time, for communication. Given that they must share it, *the effective processing time is several times higher than the physical processing time. Moreover, it increases linearly with the number of cores connected to the bus* if a single high-speed bus is used. *In vast systems, especially when attempting to mimic neuromorphic workload, the bus's speed is getting marginal.* Notice that the times shown in the figure are not proportional: the (temporal) distance between cores is in the several picoseconds range, while the bus (and the arbiter) are at a distance well above nanoseconds, so *the actual temporal behavior (and the idle time stemming from it) is much worse than the figure suggests.* This behavior is why *"The idea of using the popular shared bus to implement the communication medium is no longer acceptable, mainly due to its high contention."* [64]. For a more detailed analysis see [58], and specifically for the case of artificial neural networks [35]. The figure suggests using another design principle instead of using the bus exclusively, directly from the computing component's position.

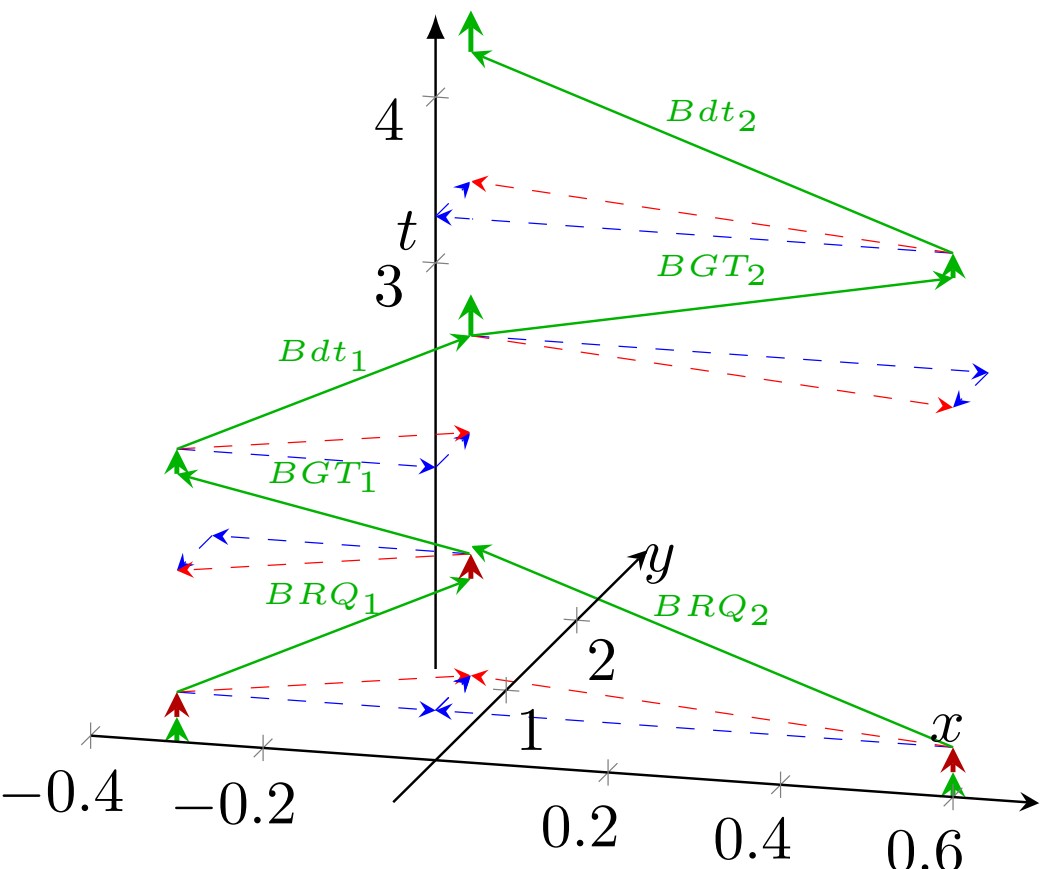

**Figure 6.** The temporal operating diagram of a technological high-speed single bus: the bus delivers data only in the fractions denoted by vertical green arrows.

*When using a shared bus, increasing either the processing speed or the communication speed, alone, does not affect linearly the total execution time anymore. Furthermore, it is not the bus speed that limits performance.* A relatively small increase in the transfer time can lead to a relatively large change in the value of the experienced processing time. This change leads to an incomprehensible slowdown of the system: its slowest component defines its efficiency. The conventional method of communication may work fine, as long as there is no competition for the bus, but leads to queuing of messages in the case of (more than one!) independent sources of communication. The effect is topped by the bursty nature of communication caused by the need for central synchronization, leads to a "communication collapse" [65], that denies huge many-processor systems, especially neuromorphic ones [66].

Notice that the issue with sharing communication resources returns in slightly different form also in quantum computing [30]: the data transfer between the quantum processor and the quantum memory needs resource sharing resemblant to the high-speed sequential bus.

### 5.5. Distributed Processing

Given that the single-processor performance stalled [6] and the building parallel computers failed [7] to reach the needed high computing performance, the computing tasks must be cut into pieces and be distributed between independently working single processors. Cutting and re-joining pieces, however, needs efforts both from programming and technology. The technology, optimized for solving single-thread tasks, hits back when several processors must cooperate, as cooperation and communication are not native features of segregated processors. The mission was so hard that the famous Gordon Bell Prize was initially awarded for achieving at least 200-fold performance gain by distributing a task between several (even thousands) processors.

Figure 7 depicts the temporal diagram of distributed parallel processing. One of the processing units (in our case, the one at (0,0.5)) orchestrates the operation, including receiving the start command and sending the result. This core makes some sequential operations (such as initializing data structures, short green arrow), then it sends the start command and operands to fellow cores at (−0.5,0) and (1,0), one after the other. Signal propagation takes time (depending on their distance from the coordinator). After that time, fellow cores can start their calculation (their part of the parallelized portion). Of course, the orchestrator Processing Unit must start all fellow Processing Units (red arrows) to begin its portion of distributed processing.

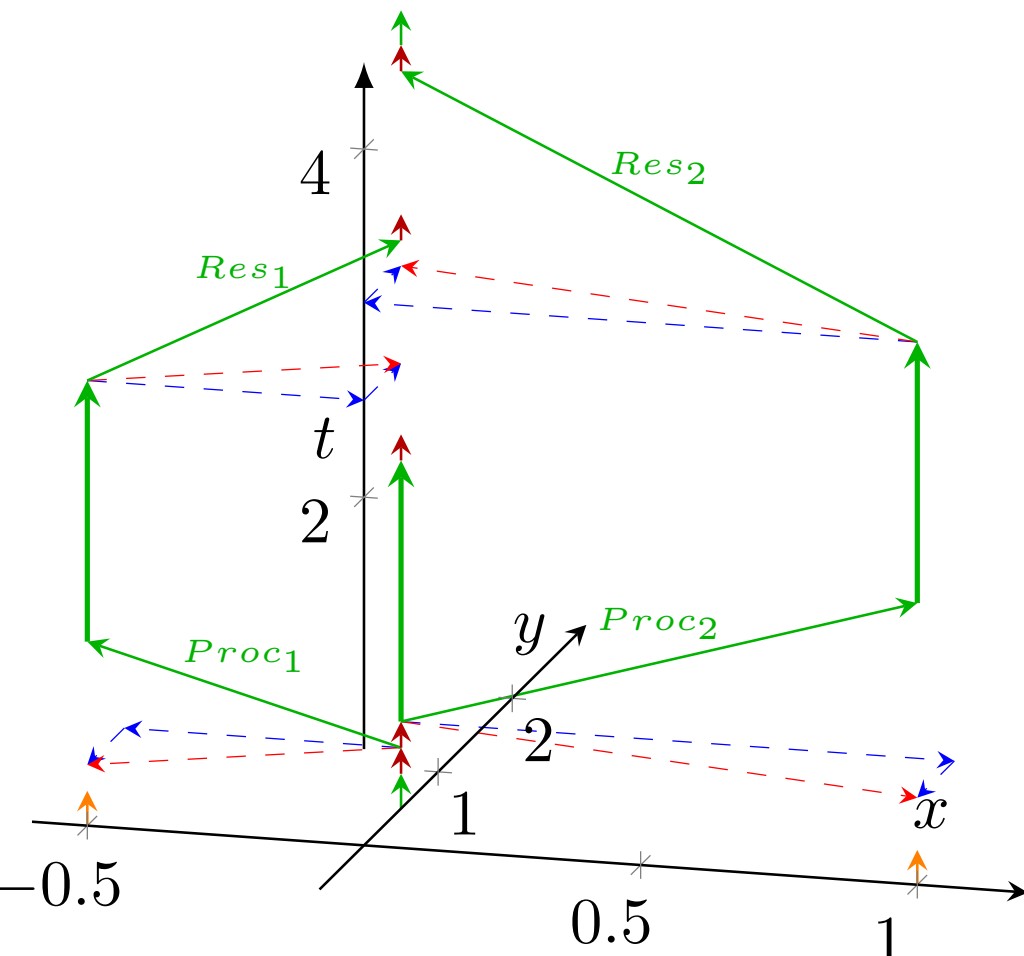

**Figure 7.** The operation of the "parallelized sequential" processing, in the time–space coordinate system. Parallel with axis *t*, *the lack of vertical arrows* signals "idle waiting" time, both for the coordinator and the fellow processors.

As the fellow processing units finish their portion, they must transmit their data $Res_i$ to the orchestrator, which receives those data in *sequential* mode, and finally makes some closing *sequential* processing. This aspect is significant; if those units' physical transmission times differ (speed or distance is different), the task is not adequately split into equal portions. Modern hardware has indeterministic behavior [67,68], or the units may be connected through an interconnection with stochastic behavior [69]. The times shown in the figure are not proportional and largely depend on the type of the system.

If the individual tasks are independent (they do not need communication), they can use dedicated processors in a Graphic Processing Unit (GPU) [70] or in a computing grid system, and can reach outstanding efficiency: they do not undergo the limitations described here. Again, the *inherently sequential-only portion* [71] of the task increases with the number

of cores and their *idle waiting time* (time delay of signals) increases with their physical size (cable length). Notice also that the orchestrating Processing Unit must wait for the results from all fellow cores, i.e., *the slowest branch defines performance.*

The dispersion in distributed systems is dramatically increased because of the physical distance of the computing components. The increase depends on the weights of critically large distances. As analyzed in [42], a different communication intensity changes the weight of the operations having considerable temporal distance. For vast distributed systems, see Figure 8, the efficiency sharply depends on the number of parallelized units and the goodness of their parallelization. "*This decay in performance is not a fault of the architecture but is dictated by the limited parallelism.*" [45] Notice how the efficiency of distributed systems also depends on the workflow they run. The estimated efficiency of brain simulation is derived from data in [54], as discussed in [42].

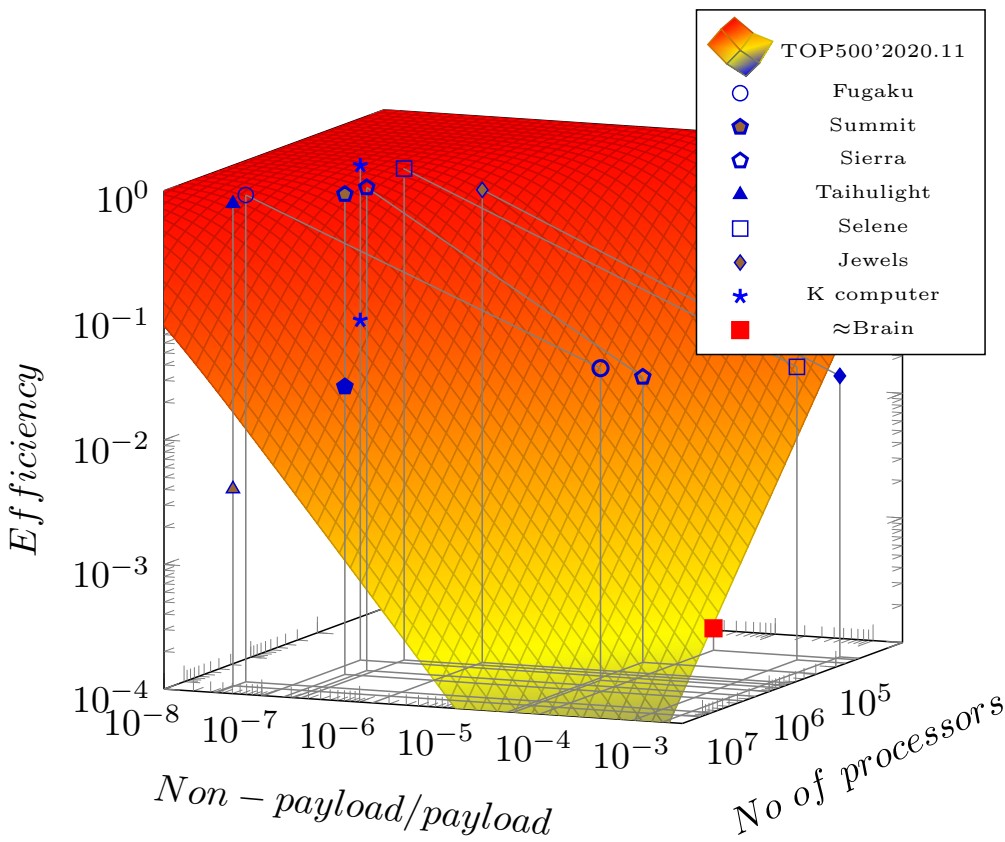

**Figure 8.** The two-parameter efficiency surface (in function of the parallelization efficiency measured by benchmark HPL and the number of the PUs) as concluded from Amdahl's Law (see [42]), in the first order approximation. Some sample efficiency values for some selected supercomputers are shown, measured with benchmarks High Performance Linpack (HPL) and High Performance Conjugate Gradients (HPCG), respectively. Moreover, the estimated efficacy of brain simulation using conventional computing is shown.

One can separate the measured efficiency values in Figure 8 into two groups. The recent trend is that only a tiny fraction of their cores are used in the HPCG benchmarking, while, of course, all their cores are used in the HPL benchmark. As discussed above, supercomputers' efficiency depends on their workload. One can project the two benchmarked efficiency values to different numbers of cores and different non-payload to payload ratios on the axes. The other group comprises measurements where the experimenters used the same number of cores in both benchmarks. For this group, for visibility, only the HPL projections are displayed.

In this latter group, the efficiency sharply decreases with the number of cores in the system. In the former group, only about 10% of the total cores is used, and the two efficiency values differ by roughly an order of magnitude. The general experience showed that the HPL-to-HPCG efficiency ratio is about 200–500 when using the same number of cores. This efficiency decrease is why these entries reduced their number of cores in the second benchmark. Their payload performance reached their "roofline" [35,72] levels at that number of cores; using all cores would decrease the system's performance by order of magnitude only because of the higher number of cores. (Started with June 2021, this "measured cores" information is missing from the published HPCG data, and even the formerly published data are removed.)

It is noticeable that the *systems having the best efficiency values do not use an accelerator*: in accelerated systems the *payload performance* gets higher but the *efficiency* is much lower in the HPL benchmark case, but it is even more disadvantageous in the case of the HPCG benchmark. As can be seen, they can reach their "roofline" efficiency with a lower number of cores; using more cores would decrease [35] their performance. In other words, the accelerators enable the systems to reach only much worse non-payload to payload ratio; furthermore, *vast accelerated systems cannot use all their cores in solving real-life tasks*.

## 6. Conclusions

The technological development made the neglections used in the commonly used computing paradigm outdated and forced us to consider the original model with correct timing relations. The stealthy nature of the development led to many technological implementations, including the synchronous operation and the high-speed bus, which is not usable anymore. One obvious sign is the enormous dissipation of our computing systems, which is a direct consequence of the computing systems' dispersion being well above the theoretically acceptable level. "Reinventing electronics" [20] is a must, and not only for building neuromorphic computing devices. The computing model is perfect, but the classic paradigm is used far outside of its validity range. For today's technological conditions, the needed "contract" [7] between mathematics and electronics should be based on a paradigm that considers the transfer time. Our findings show that we can build a generalization of the classic computing paradigm based on the idea of the finite interaction speed between computing units. The classic computing paradigm remains usable for simple, low-speed, and low communication systems. Still, for modern processors with a vast number of transistors, supercomputers having an excessive size, biological systems with low interaction speed, only the generalized paradigm provides a good description. The generalization enables explaining issues with present-day computing, from high power consumption to low computing performance, and suggests ideas for improving technology to enhance computing.

**Funding:** Project no. 136496 has been implemented with the support provided from the National Research, Development and Innovation Fund of Hungary, financed under the K funding scheme.

**Institutional Review Board Statement:** Not applicable.

**Informed Consent Statement:** Not applicable.

**Data Availability Statement:** Not applicable.

**Conflicts of Interest:** The author declares no conflict of interest.

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
