# Peer review of "Revising the Classic Computing Paradigm and Its Technological Implementations"

_informatics, doi:10.3390/informatics8040071_

Round 1

Reviewer 1 Report

This is an interesting article based on the analysis of the model introduced by Von Neumann. The review of the classic paradigm is performed accurately.

However, I would like to point out the following concerns:

  • the Research question is not clearly identified in the abstract and in the introduction
  • The methodology used (review? cases analysis?) should be separately disclosed in a specific paragraph after the introduction
  • alternative models such as quantum computing, neural networks, Turing Machines are not even mentioned although the article seems to present the Von Neumann architecture as the leading one in the future (lines 43-44). This is questionable, considering, in particular, the recent, increasing advances in quantum computing. The author(s) is(are) expected to justify this assumption and to compare the model with the others in a specific section if he/she/they want to claim the supremacy of this model.
  • some references are expected at the end of lines 101 and 118.
  • The conclusions must be developed and integrated with a proper discussion, presenting the findings, identifying the limitations of the research, supporting the assumptions.

Author Response

This is an interesting article based on the analysis of the model introduced by Von Neumann.
The review of the classic paradigm is performed accurately.

Thank you.

In the manuscript, the replies are annotated, using the 'track changes' option, as R1/Qx, with reference to the questions below. Given that that questions are eventually relied at different points in the MS, and there is a large number of changes, the replies are not repeated entirely here.

However, I would like to point out the following concerns:

 +Q1:   the Research question is not clearly identified in the abstract and in the introduction
 Fixed:

Two sentences patched in the abstract and two paragraphs added in the introduction.

 +Q2:   The methodology used (review? cases analysis?) should be separately disclosed
    in a specific paragraph after the introduction
    Fixed.

Originally, the MS was intended to be a research paper, but, thanks to reviewers question, not it is a review.

+Q3:   alternative models such as quantum computing, neural networks, Turing Machines are not even mentioned
       although the article seems to present the Von Neumann architecture as the leading one in the future
       (lines 43-44).
       Fixed:

I pinpointed the term 'model' as used in the MS, and added subsubsections to subsection 3.5, discussing the two mentioned fields.

+ Q4:   This is questionable, considering, in particular, the recent, increasing advances in quantum computing.
       The author(s) is(are) expected to justify this assumption
       and to compare the model with the others in a specific section
       if he/she/they want to claim the supremacy of this model.
       Fixed:

In subsection 3.5, the general usability of quantum computing and ANNs is discussed and evaluated.

 +Q5:   some references are expected at the end of lines 101 and 118.
 Fixed

It was missing; added

 +Q6:   The conclusions must be developed and integrated with a proper discussion,
 presenting the findings, identifying the limitations of the research, supporting the assumptions.
 Fixed

Both the body of the MS and the Summary extended.

Reviewer 2 Report

The paper describes a review about the classic computing paradigm and its technological implementations.

Despite the huge effort of the author, the paper requires of very improvement.

Into introduction section the word "von Neumann" appears many times.

Same problem is present from lines 48-59.

From lines 63- 66 insert comma into itemize.

Figure 1 needs to be explained.

At line 152 it is not necessary insert sentence into brackets.

Lines 158-165 need to be extended by writing and explaining the word EDVAC.

The four-dimensional system requires to be shown in mathematics form.

Also lines 331-338 require a mathematical modeling.

Figure 5 must be clarified. 

Listing 1 requires to be explained after that it is shown.

Figure 6 must be inserted into subsection 5.4

The references are poor for a review article.

I suggest the following references both for recent parallel architectures and. old machines:

Mariantoni, M., Wang, H., Yamamoto, T., Neeley, M., Bialczak, R. C., Chen, Y., ... & Martinis, J. M. (2011). Implementing the quantum von Neumann architecture with superconducting circuits. Science, 334(6052), 61-65.

Fiscale, S., De Luca, P., Inno, L., Marcellino, L., Galletti, A., Rotundi, A., ... & Quintana, E. (2021, June). A GPU Algorithm for Outliers Detection in TESS Light Curves. In International Conference on Computational Science (pp. 420-432). Springer, Cham.

Poznanovic, D. S. (2006, March). The emergence of non-von neumann processors. In International Workshop on Applied Reconfigurable Computing (pp. 243-254). Springer, Berlin, Heidelberg.

Birkhoff, G., & Von Neumann, J. (1936). The logic of quantum mechanics. Annals of mathematics, 823-843.

Some table is necessary to highlight the difference of past and present architectures.

Conclusion must be improved.

Author Response

   Despite the huge effort of the author, the paper requires of very improvement.

Many thanks for the inspiring questions and suggestions; also for the suggested cited references.

The MS is submitted in form of 'track changes', where the changes are annotated as R2/Qx, where x refers to the questions below. Given that a large amount of changes was carried out, and some questions are answered at multiple points in the MS, the detailed answers are not repeated here.

+Q1: Into introduction section the word "von Neumann" appears many times.
Fixed

+Q2: Same problem is present from lines 48-59.
Fixed

+Q3: From lines 63- 66 insert comma into itemize.
Fixed

+Q4: Figure 1 needs to be explained.
Fixed

An extra paragraph added.

+Q5: At line 152 it is not necessary insert sentence into brackets.
Fixed

+Q6: Lines 158-165 need to be extended by writing and explaining the word EDVAC.
Fixed. A complete citation added.

+Q7: The four-dimensional system requires to be shown in mathematics form.
Fixed. A textbook citations added.

+Q8: Also lines 331-338 require a mathematical modeling.
Fixed

+Q9: Figure 5 must be clarified.
Fixed. A textbook citations added, plus some extra sentences added.

+Q10: Listing 1 requires to be explained after that it is shown.
Fixed. Three extra sentences added.

+Q11: Figure 6 must be inserted into subsection 5.4
Fixed. It is a floating figure, and the technical editor may still move it.

+Q12: The references are poor for a review article.
The paper was originally a research paper, but thanks for the valuable references.
They are built in the revised MS, and several review papers are cited.

Round 2

Reviewer 2 Report

All suggested comments have been addressed. Now the paper has a very good structure in order to be processed for the publication.